# Infants' brain activity to cartoon face using functional near-infrared spectroscopy

**Nanako Yamanaka**[1]*, **So Kanazawa**[2], **Masami K. Yamaguchi**[1]

**1** Department of Psychology, Chuo University, Tokyo, Japan, **2** Department of Psychology, Japan Women's University, Kanagawa, Japan

* a17.cnay@g.chuo-u.ac.jp

## Abstract

In this study, to investigate whether infants showed face-specific brain activity to a cartoon human face, we conducted a functional near-infrared spectroscopy (fNIRS) experiment and a behavioral experiment. In the fNIRS experiment, we measured the hemodynamic responses of 5- and 6-month-old infants to cartoon female and cartoon character faces using fNIRS. The results showed that the concentration of oxy-Hb increased for cartoon female faces but not for cartoon character faces. This indicates that face-specific brain activity occurred for cartoon female faces but not cartoon character faces, despite the fact that both are faces. In the behavioral experiment, we examined whether the 5- and 6-month-old infants preferred cartoon female faces to cartoon character faces in the upright and inverted conditions. The results showed a preference for cartoon female faces in the upright but not in the inverted condition. This indicates that 5- and 6-month-old infants can perceive cartoon female faces, but not cartoon character faces, as faces. The results of the two experiments indicated that face-specific brain activity occurred for cartoon female faces. This indicates that infants can perceive cartoon female faces as faces.

## Introduction

Humans have the ability to perceive a face even in non-realistic faces, such as in cartoon faces used in animation. A previous study reported that infants can perceive a cartoon face as a face in behavioral experiments [1]. Kobayashi et al. recently investigated whether infants perceive a cartoon face as a face [1]. They compared the infants' preference between the mother's and a stranger's face of the cartoon image and found that infants preferred the mother's face even in the cartoon image. The results suggest that infants can recognize the mother's face in a cartoon image. This study indicated infants' ability to perceive cartoon face processing in behavioral experiments, although no prior study has examined infants' brain activity in response to cartoon faces. In this study, we investigated whether cartoon faces induced brain activity similar to photographed faces using functional near-infrared spectroscopy (fNIRS).

Previous studies have shown that infants can recognize various face types, such as cartoon faces and faces applied global liner transformation as faces [2], and that the composite face effect occurs in infants [3]. Yamashita et al. investigated the effect of global linear

**Data Availability Statement:** All relevant data are within the paper and its Supporting Information files.

**Funding:** This research was financially supported by "Construction of the Face-Body Studies in

Transcultural Conditions" (17H06343). The funders had no role in study design, data collection and analysis, decision to publish, or preparation of the manuscript.

**Competing interests:** The authors have declared that no competing interests exist.

transformations on the recognition of mothers' faces in infants [2]. They applied global linear transformations (shearing, horizontal stretching, and vertical stretching) to both the mother's face and the stranger's face, and tested infants' preferences between these faces. 7-month-old infants showed a preference for the mother's face with vertical stretching but not with shearing and horizontal stretching. These findings suggest that 7-month-old infants might recognize the mother's face even during vertical stretching, similar to adults. Nakato et al. examined whether the composite face effect occurred in infants aged 5-8months [3]. The composite face effect is that participants are slower and less accurate in recognizing the top half of one face presented in a composite with the bottom half of another face. This indicates the holistic processing of the face. In their study, they compared infants' preference for the mother's face between the composite and non-composite face conditions. The results showed that 7- to 8-month-old infants preferred the mother's face in the non-composite condition but not in the composite condition. That is, infants can recognize the mother's face in non-composite but not composite conditions. This indicates that the composite face effect occurred only in 7- to 8- month-old infants. This suggests that infants older than 7 months are able to process familiar faces holistically. Recently, Kobayashi et al. investigated the development of the ability to recognize familiar faces in drawings [1]. They examined the preference for the mother's face between photographed and cartoon images in 6- to 8-month-old infants, and only 7- to 8-month-old infants showed a preference for mother's faces regardless of a photographed or cartoon image. The results suggested that cartoon faces as well as a photographed faces could be recognized as faces by 7- to 8-month-old infants. These previous studies revealed that 7-month-old infants showed holistic face processing for various face types.

In adults, face-specific brain activities in various face types have been investigated [4–7]. Tong et al. examined brain activity in response to a variety of face-like stimuli such as cartoons and cat faces using functional magnetic resonance imaging (fMRI) [4]. Cartoon and cat faces as well as human faces induced fusiform face area (FFA) responses. This result suggests a generalization of the FFA response across different face types. Gomez et al. indicated that extensive experience with cartoon characters induced face-specific brain activity [5]. They investigated whether Pokémon induced face-specific activities through extensive childhood experiences. They measured the brain activity in response to Pokémon in adults with and without childhood experience of playing Pokémon. The responses to Pokémon in the occipito-temporal sulcus (OTS) were shown only in participants with Pokémon experience. The face-specific brain activity in a boy with autism who had a special interest in Digimon was investigated by Grelotti et al. [6]. They compared brain activity in response to Digimon and to the human face using fMRI and showed that his amygdala and fusiform gyrus were activated by Digimon but not the human face. The event-related potential (ERP) study also suggested face-specific brain activity to a variety of face-like stimuli. Sagiv et al. investigated whether face-specific brain activity was induced by photographs of natural faces, realistically painted portraits, sketches of faces, and schematic faces using ERPs [7]. They compared the N170 ERP components elicited by these faces, and N170 did not distinguish between different face types. These results suggest that even the schematic face made from simple line fragments triggered the N170.

In this study, we aimed to investigate face-specific brain activity in response to cartoon faces in infants using fNIRS. Kobayashi et al. have demonstrated that 7- to 8-month-old infants show a preference for the mother's face, regardless of whether these faces are presented as photographed or cartoon images [1]. This result indicates that 7- to 8-month-olds can perceive cartoon faces as faces. In this study, we measured the hemodynamic responses of 5- to 6-month-old infants to cartoon female faces and cartoon character faces using fNIRS. We examined the differences in face-specific brain activity in response to cartoon female faces and

cartoon character faces. We predicted that infants could perceive cartoon female faces as faces [1] but not cartoon character faces. In this case, we hypothesized that face-specific hemodynamic responses, as in previous studies [8–11], should be induced by cartoon female faces but not cartoon character faces. Before the fNIRS experiments, we conducted behavioral experiments to examine whether infants prefer cartoon female faces to cartoon character faces using the preferential looking method. In two orientation conditions (upright and inverted), we examined whether infants looked longer at the cartoon female face than at the cartoon character face. In general, the recognition of an inverted face is poor compared to an upright face. This effect, known as the face-inversion effect, occurs only on the face. An infant study showed that this effect exists in 4-month-old infants [12]. We predicted the infants' preference for cartoon female faces in the upright but not in the inverted condition.

## Experiment 1

### Materials and methods

**Participants.**   38 healthy 5- and 6-month-old infants (17 boys and 21 girls, mean age of 164.71 days, range from 136–193 days) participated in the experiments. Infants were assigned randomly and in equal numbers to the orientation conditions (upright and inverted). An additional 13 infants were tested but excluded from the final analysis owing to fussiness or an insufficient number of available trials (fewer than 30 s for both sides of the stimuli). The infants were recruited through newspaper advertisements. This study was conducted according to the Declaration Helsinki and was approved by the Ethical Committee of Chuo University. Parents gave prior written informed consent for their children's participation and for publication of the results in an online open-access publication.

**Stimuli.**   Stimuli were color images of three cartoon human female faces and three cartoon character faces (Fig 1). All cartoon female faces were automatically produced using free software available online (http://www.photo-kako.com/) by uploading the original images. These faces were in the frontal view with neutral expression. The cartoon female face stimuli were 13.3˚×17.9˚ in size. The cartoon character face stimuli were used the characters of "ANPAN-MAN" that is the famous Japanese cartoon. The cartoon character face stimuli were 14.5˚×16.0˚ in size. The inverted cartoon female and cartoon character faces were produced by rotating the original upright face by 180˚.

**Apparatus.**   All infants were tested while sitting on their parent's lap at a viewing distance of approximately 40 cm from a 24-inch LCD monitor, which was controlled by a computer. Both the infant and the monitor were surrounded by an enclosure covered with black cloth. Each infant's looking behavior throughout the experiment was recorded through a charge-couple device (CCD) camera positioned directly below the monitor, which was connected to a TV monitor and a digital video recorder positioned outside the enclosure. The experimenter started the sequence of the trial by looking at the live image of the infant's face displayed on the TV monitor. The recorded images of the infant's face and looking behavior allowed for the offline coding of looking times.

**Procedure.**   Infants were randomly assigned to one of two orientation conditions (upright and inverted). The preferential looking method was used to measure each infant's response. The experimental session consisted of six 10 s trials. At the beginning of each trial, a cartoon with a brief sound was presented at the center of the monitor to attract the infant's attention. The experimenter initiated each trial as soon as the infant began paying attention to the cartoon. In each trial, a cartoon female face and a cartoon character face were presented side by side on the monitor. The position and pair of stimuli were counterbalanced across the infants.

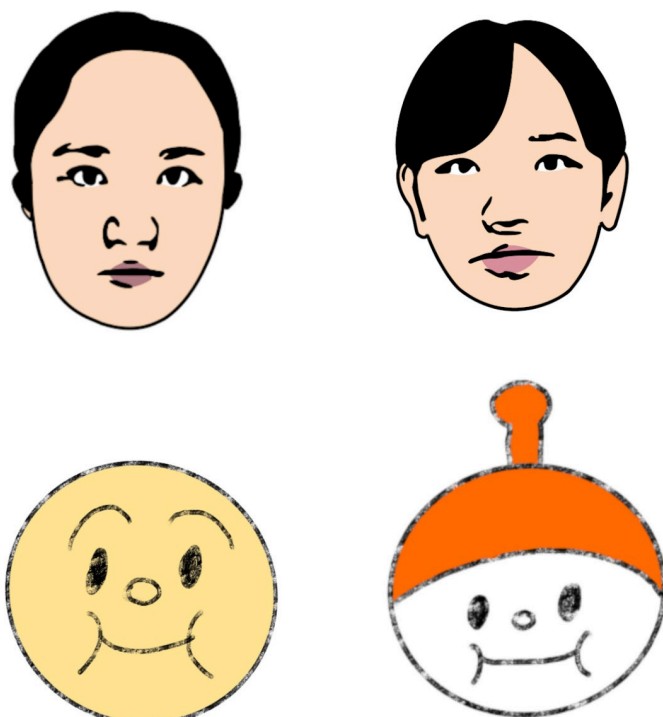

**Fig 1. Examples of the cartoon female and character faces used as stimuli.** Faces were presented on a gray background. The cartoon character face stimuli used in the experiment were different from the images in Fig 1. Because of the copyright, we used other character faces here, instead of the "ANPANMAN." For details of the cartoon character face stimuli, see the Materials and Methods section in Experiment 1.

One observer analyzed frame-by-frame video recordings of infants' eye movements to record the total looking time at each of the two faces during each phase. In addition, video recordings of 5 infants were analyzed by a second independent observer. Both observers were unaware of the position (left/right), where the cartoon female and cartoon character faces would be presented on the screen. Interobserver reliability throughout the experiment was high (Pearson's $r$ = .88, for the duration of fixation).

## Result

We calculated an individual percentage preference score for cartoon female faces over cartoon character faces in two orientation conditions (upright/inverted). The mean preference score is shown in Fig 2. The preference scores were calculated by dividing the infants spent looking at the cartoon female faces during the 6 test trials by their total looking time over 6 test trials. The score obtained was multiplied by 100. A preference for cartoon female faces indicates above chance (50%), and a preference for a cartoon character face indicates below chance. If the infants detected faces in the cartoon female faces, they would look longer at them than at the cartoon character faces in the upright but not inverted condition. To test whether the infants prefer the cartoon female faces, we conducted a two-tailed $t$-test on the preference score (vs. a chance level of 50%). We found that 5- and 6-month-old infants significantly preferred cartoon female faces to cartoon character faces in the upright condition ($t$ (18) = 2.80, $p$ < .05, $d$ = 0.93), but not in the inverted condition ($t$ (18) = 1.13, $ns$.).

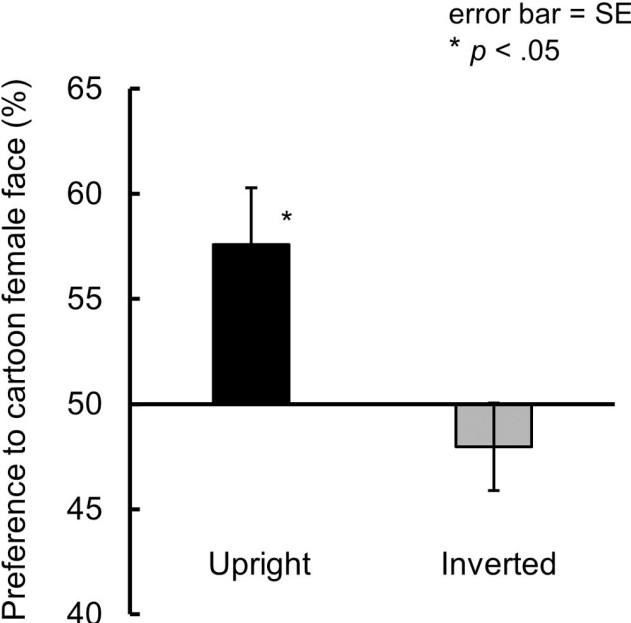

error bar = SE
* *p* < .05

**Fig 2. Preference scores for cartoon female face (vs. cartoon character face) in upright and inverted conditions (Experiment 1).** A preference for a cartoon female face indicates above chance (50%), and a preference for cartoon character face indicates below chance. Error bars represent standard error.

## Experiment 2

### Materials and methods

**Participants.** 11 healthy 5- and 6-month-old infants (6 boys and 5 girls, mean age of 180.72 days, range from 155–191 days) participated in the experiments. An additional 11 infants were tested but excluded from the final analysis because of an insufficient number of available trials (fewer than three trials for either the cartoon female face or cartoon character face condition) owing to crying, failure to look at stimuli, or motion artifacts. Sample size was based on previous studies testing infants' cortical hemodynamic changes during observation of face stimuli [8, 13, 14]. The infants were recruited through newspaper advertisements. This study was conducted according to the Declaration Helsinki and was approved by the Ethical Committee of Chuo University. Parents gave prior written informed consent for their children's participation and for publication of the results in an online open-access publication.

**Stimuli and design.** The stimulus presentation consisted of a test and a baseline period (Fig 3). The stimuli for the test period were color images of five cartoon female faces and five cartoon character faces. The stimuli for the baseline period were color images of five cartoon vegetables. Cartoon vegetables were automatically produced using free software available online (http://www.photo-kako.com/) by uploading the original images. The cartoon female face stimuli were 17.1˚×23.3˚, the cartoon character face stimuli were 19.4˚×21.4˚, and the vegetable stimuli were 16.8˚×16.8˚ in size.

In each trial, five faces were presented in random order at a rate of 1 Hz under either the cartoon female or cartoon character condition. The faces of cartoon females and cartoon characters were presented in alternating trials. Faces of cartoon females were shown in half of the trials and those of cartoon characters in the other half. The order of the two conditions was counterbalanced across the infants. The total duration of each test period was fixed at 5 s.

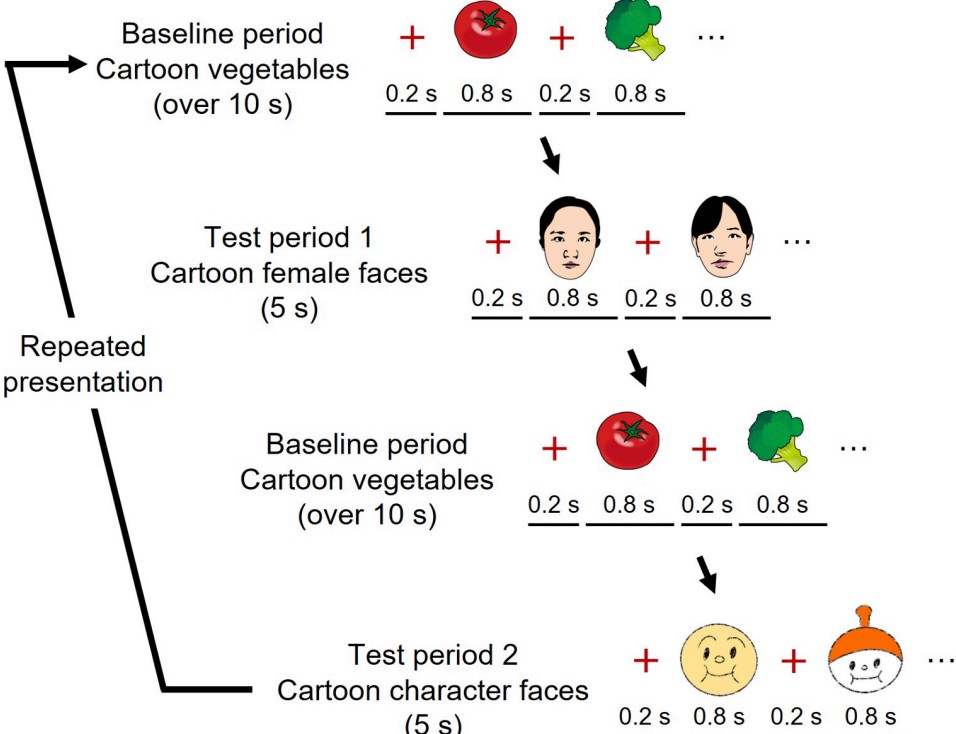

**Fig 3. Experimental procedure.** In each trial, the baseline period consisted of the presentation of the images of five cartoon vegetables. The duration of the baseline period was at least 10 s. The test period consisted of the presentation the images of five cartoon female faces and cartoon character faces. The duration of the test period was 5 s. The presentation order was altered for test periods 1 and 2 for each infant. The cartoon character face stimuli used in the experiment were different from the images in Fig 3. Because of the copyright, we used other character faces here, instead of the "ANPANMAN".

Each test period followed a baseline period. During the baseline period, five vegetables were shown in a random order at a rate of 1 Hz. The baseline period was controlled by the experimenter, and the duration was at least 10 s. The results obtained from viewing vegetables were used as the baseline.

In both the baseline and test periods, the stimulus duration was 800 ms, and a small red cross was presented during the 200 ms interstimulus interval. To attract and retain the infant's attention, both the face and vegetable stimuli were accompanied by a beeping sound presented at 1 Hz. Two different sounds were used for the face stimuli and the vegetables, but the same sound was used for the cartoon female and cartoon character faces.

**Apparatus and procedure.** All stimuli were displayed on a 32-inch LCD monitor (Display++, Cambridge Research Systems [CRS]) with a resolution of 1920×1080 pixels controlled by a computer. Both the infant and LCD monitor were located inside an enclosure made of iron poles and covered with black cloth. The infant's viewing distance from the LCD monitor was approximately 60 cm. There were two loudspeakers, one on either side of the LCD monitor. A CCD camera was placed directly below the monitor screen. Throughout the experiment, the infant's behavior was videotaped using this camera. The experimenter could observe the infant's behavior via a video monitor connected to the CCD camera.

Each infant was tested while sitting on their parent's lap and facing a computer screen. The infants watched the stimuli passively while their brain activity was measured, and they were allowed to watch the stimuli for as long as they were willing to.

**Recoding.** We used a Hitachi ETG-4000 system (Hitachi Medical, Chiba, Japan) to measure hemodynamic changes in oxy-Hb and deoxy-Hb concentrations using 24 channels with a 0.1 s time resolution. Twelve channels each were assigned for measurements in the right temporal and the left temporal areas. Two wavelengths of near-infrared light (695 and 830 nm) were projected through the skull.

The fNIRS probes (Hitachi Medical) contained nine optical fibers (3×3 arrays) comprising five emitters and four detectors. The optical fibers were kept in place using a soft silicon holder. The distance between the emitters and detectors was set to 2 cm. Each pair of adjacent emitting and detecting fibers is defined as a single measurement channel.

The probes were placed over the same locations on the bilateral temporal areas centered at T5 and T6 according to the International 10–20 system [15] (see [9, 10, 13, 16–18], for other infant studies recoding from the same sites) (Fig 4).

When the probes were positioned, the experimenter checked to see whether the fibers were touching each infant's scalp correctly. The Hitachi ETG-4000 system automatically detects whether the contact is adequate to measure the emerging photons for each channel. The channels were excluded from the analysis if adequate contact between the fibers and the infant's scalp could not be achieved because of interference by hair.

**Data analysis.** Before conducting the data analysis, we determined the valid trials used for the statistical analysis. We excluded trials (1) when the infants did not look at the test stimuli for 5 s or became fussy, (2) when they looked back at the face of their parent's face during the preceding baseline period, or (3) when movement artifacts were detected. Criteria (1) and (2) were detected by examining the infant's behavior that recorded on the video. Criterion (3) was detected by the analysis of shape changes in the time series of raw data of oxy-Hb concentration.

The changes in oxy-Hb and deoxy-Hb concentrations from individual channels were digitally bandpass-filtered at 0.02–2.0 Hz to eliminate longitudinal signal drift and high frequency noise owing to heartbeat pulsations [19]. We then averaged the oxy-Hb and deoxy-Hb changes in each channel across the trials from 1s before the test period onset to 1 s after the test period,

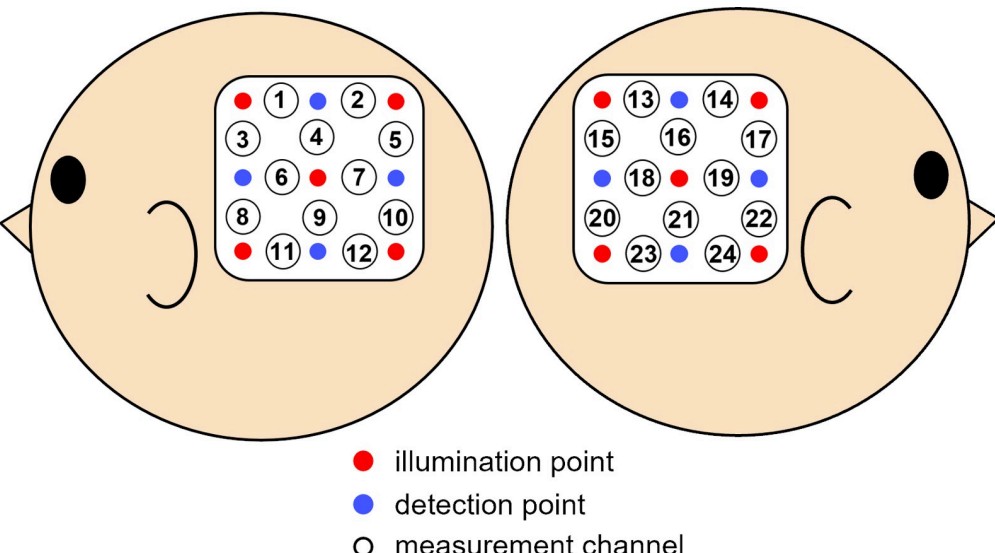

**Fig 4. Locations of the measurement channels.** The probes were placed on the left and right temporal areas centering at T5 and T6 of the international 10–20 system. The blue circles represent for detector, and the red circles represent for emitter. Each number correspond to the measurement channels.

offset for each participant in a time series of 0.1 s time resolutions [9]. To normalize the mean concentrations in the time series of oxy-Hb and deoxy-Hb data, we calculated Z-scores at each time point during the cartoon female face and cartoon character face condition, for each channel of each participant. The Z-score ($d$) was calculated by subtracting the mean concentration of the baseline period ($m_{baseline}$) from the concentration at each time point during the test period ($m_{test}$), which was divided by the standard deviation of the concentration data during the baseline period ($sd$), as follows:

$$d = (m_{test} - m_{baseline})/sd$$

The 1 s immediately before the onset of each test trial was defined as baseline. The concentration changes measured by fNIRS were originally relative values and could not be averaged directly across participants. Normalized data such as the Z-scores, however, could be averaged regardless of the unit [20, 21].

Statistical analyses were conducted on these mean Z-scores for the oxy- and deoxy-Hb concentrations. To examine the channel showed greater activation for the cartoon female and cartoon character face comparing to the object (baseline), two-tailed one-sample $t$-tests vs. a chance level of 0 (baseline) were conducted for the mean Z-score in each of the 24 channels. Furthermore, to investigate the difference between presentation of the cartoon female and cartoon character face condition, we conducted two-tailed paired $t$-tests on the mean Z-scores.

Previous studies have shown that the infants' hemodynamic responses typically lags behind stimulation and peak approximately 8–10 s after stimulus presentation [22–24]. This implies that, the dynamic response would reach peak response after the end of the 5-s presentation of test stimuli in the current study. However, in a pilot study that measured the hemodynamic response for a longer period, the data obtained after the presentation of the test stimulus were very noisy due to the infants' body movements and, therefore they, were not suitable for analysis. Moreover, in several trials, we observed that many infants looked away from the LCD monitor immediately after the test trial offset and then looked back at the monitor again within a few seconds. Based on this observation, we performed statistical analysis against the mean Z-score of the last 1 s of the test trials to avoid data that included motion artifacts.

## Result

We obtained hemodynamic responses from ten 5- to 6-month-old infants who looked at the stimuli for more than three trials on both the cartoon female and cartoon character face conditions. The mean number of valid trials was 4.18 (SD = 1.11) for cartoon female face condition and 3.82 (SD = 1.03) for cartoon character face condition. There were no significant differences on the mean number of trials between the two conditions ($p > .05$, two-tailed).

Fig 5 shows the time course of average oxy-Hb concentration during the cartoon female face condition (the blue line) and the cartoon character face condition (the red line) in each of the 24 channels. We examined the channels activated in response to the cartoon female face condition and cartoon character face condition compared to baseline of 0. For each of the 24 channels, two-tailed one sample $t$-tests vs. a baseline of 0 were conducted for the mean Z-score of oxy-Hb and deoxy-Hb. Channel 23 showed a significant increase in the concentration of oxy-Hb ($t$ (10) = 4.17, $p < .01$ after Bonferroni's correction, $d = 1.32$). There were no channels showing significant changes in the concentration of deoxy-Hb (all $ps > .10$ after Bonferroni's corrections). In the cartoon character face condition, there were no channels showing significant changes in the concentrations of oxy-Hb and deoxy-Hb (all $ps > .10$ after Bonferroni's corrections). The greater activation compared to the baseline was found in the cartoon female face condition only in the channel 23.

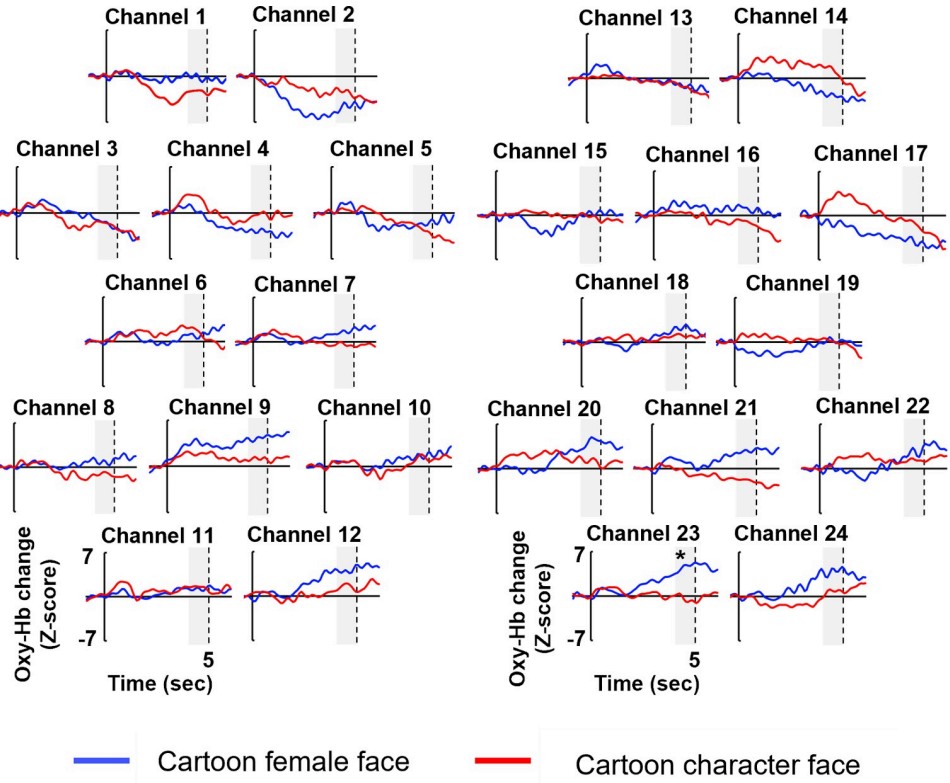

**Fig 5. The time course of average oxy-Hb concentration during the cartoon female and cartoon character face conditions in each of the 24 channels.** The blue line in the graph represents the mean Z score in the cartoon female face condition, and the red line represents that in the cartoon character face condition. On the horizontal axis, 0 represents the beginning of the test period and 5 represents the end of the test period. The shaded area represents the zone for the statical analysis, the last 1 s of the test period. In 23 channel, the concentration of oxy-Hb in cartoon female face condition were significantly greater than chance level of 0 and that in cartoon character face condition. $^*p < 0.05$.

To investigate whether there is a significant difference in oxy- and deoxy-HB concentrations between presentation of cartoon female and cartoon character face condition in channel 23, we conducted a two-tailed paired *t*-test. We found that the concentration of oxy-Hb showed a greater increase in response to the cartoon female face condition than in response to the cartoon character face condition ($t$ (10) = 6.15, $p < .01$, $d = 1.94$). There were no significant differences in deoxy-Hb concentration between presentation of cartoon female and cartoon character face condition in channel 23 ($t$ (10) = 1.08, $p > .10$).

## Discussion

This study investigated the face-specific activity to cartoon faces in infants. We examined the hemodynamic responses to cartoon female faces and cartoon character faces using fNIRS. Before conducting fNIRS experiments, we conducted behavioral experiments to examine whether infants prefer cartoon females to cartoon character faces using the preferential looking method.

The results of a behavioral experiment (Experiment 1) revealed that the 5- and 6-month-old infants preferred cartoon female faces in the upright but not inverted condition. The result of the fNIRS experiment (Experiment 2) revealed that the 5- and 6-month-old infants showed increased concentration of oxy-Hb for cartoon female faces but not for cartoon character faces

in channel 23. These results mean that 5- and 6-month-old infants can perceive cartoon female faces, but not cartoon character face, as faces.

Since 2003, developmental studies using fNIRS in infants have revealed that it can detect cortical activation induced by various visual patterns [24], speech [25], objects [26], and faces [27]. Since 2007, fNIRS studies have reported that the temporal region of the brain is activated by faces in infants. Otsuka et al. investigated brain activity in response to upright and inverted faces in 5- to 8-month-old infants using fNIRS [8]. They examined the hemodynamic responses to upright and inverted faces and showed that the concentration of oxy-Hb and total-Hb increased significantly in the right lateral area during the upright face condition. Lloyd-Fox et al. examined social brain network activity in 5-month-old infants using fNIRS [28]. They measured the hemodynamic responses to faces and objects in the temporal regions and showed that the concentration of oxy-Hb increased only for faces. Nakato et al. investigated the brain activity of 5- and 8-month-old infants in response to frontal and profile views using fNIRS [9]. They found that the concentration of oxy-Hb and total-Hb in the right temporal regions increased only for frontal views in 5-month-old infants. The 8-month-old infants showed that the concentration of oxy-Hb and total-Hb in the right temporal regions increased for both frontal and profile views. These fNIRS studies suggest that the temporal region is specifically activated by faces, even in infants.

Previous studies have revealed that infants can perceive face-like objects as faces. Even newborns can detect face-like patterns. Valenza et al. investigated the ability of newborns to detect face-like patterns [29]. They showed that newborns preferred face-like patterns composed of elements with a face-like configuration over non-face-like patterns composed of elements with a non-face-like configuration. 3- and 4-month-old infants could detect a face in the two-tone images referred to as Mooney faces. Otsuka et al. showed that infants preferred upright over inverted Mooney faces [30]. 7- and 8-month-old infants could detect a face in Arcimboldo images [13]. Arcimboldo images are an imaginative portrait composed of fruits and vegetables. They investigated whether infants recognize a face in Arcimboldo images using the preferential looking technique and fNIRS. They showed that 7- and 8-month-old infants preferred upright to inverted Arcimboldo images. Additionally, they showed that the concentration of oxy-Hb increased in the left temporal area during the presentation of the upright Arcimboldo images in these infants. These studies compared the infants' preference between a face configuration and a non-face configuration; however, no studies have investigated infants' preferences between a human face and a non-human face. Our study is the first to compare the infants' face-specific brain activity in response to a human and a non-human face, that is, a cartoon female face and a cartoon character face.

Grelotti et al. investigated brain activity in response to the human face and Digimon in two boys with autism using fMRI [6]. In an 11-year-old boy who had a special interest in Digimon, brain activity in his fusiform gyrus was induced by Digimon but not the human face. In a 17-year-old boy who did not have a special interest in Digimon, brain activity in his fusiform gyrus was not induced by either Digimon or the human face. According to their study, the human face could not induce activity of the FFA in either an 11-year-old or a 17-year-old boy with autism. The current study shows that the occipito-temporal regions of the 5- to 6-month-old infants activated for the cartoon female face, but not for the cartoon character face. To summarize the two studies, the human face could induce activity in typically developing infants, but not in juveniles with autism. Future studies could investigate abnormal brain activity in response to the human face in 5- to 6-month-old infants at risk due to being younger siblings of children with autism. This task may aid in the diagnosis of autism in the early infant period.

In our study, channel 23 could discriminate between the cartoon female and cartoon character face. In other words, the 5- and 6-month-old infants showed an increase in oxy-Hb

concentration for the cartoon female face but not for the cartoon character face in channel 23. Previous fNIRS studies have also been suggested that channel 23 is activated for faces [9, 17]. Nakato et al. found that channel 23 showed an increase in oxy-Hb concentration for profile-view face presentations in 8-month-old infants [9]. Kobayashi et al. found that channel 23 showed a reduction in the concentration of oxy-Hb for the adaptation of identical faces in 5- to 6-month-old infants [17]. These studies revealed that channel 23 was involved in the perception of profile view and the representation of facial identity. In our study, we found that channel 23 showed an increase in oxy-Hb concentration for the cartoon female face. Our study indicates that the cartoon female face, but not the photographed female face, could induce activity in channel 23.

This study investigated whether infants showed face-specific brain activity in response to the cartoon face. We measured the hemodynamic responses of 5- and 6-month-old infants to cartoon female faces and cartoon character faces using fNIRS. The results showed that the concentration of oxy-Hb increased for cartoon female faces but not for cartoon character faces. Before conducting fNIRS experiments, we conducted behavioral experiments to examine whether these infants preferred cartoon female faces to cartoon character faces in the upright and inverted conditions. The results showed a preference for cartoon female faces in the upright but not inverted condition. To conclude, 5- and 6-month-old infants can perceive cartoon female faces, but not cartoon character faces, as faces. According to Gomez et al. [5], infants' future extensive experience with cartoon character faces might induce face-specific brain activity.

## Supporting information

**S1 Table. Individual infants' preference scores for cartoon female faces in the upright and inverted conditions (Experiment 1).**
(TIF)

**S2 Table. Individual infants' Z-scores of oxy-Hb for cartoon female faces in each of channels (Experiment 2).**
(TIF)

**S3 Table. Individual infants' Z-scores of oxy-Hb for cartoon character face faces in each of channels (Experiment 2).**
(TIF)

**S4 Table. Individual infants' Z-scores of deoxy-Hb for cartoon female faces in each of channels (Experiment 2).**
(TIF)

**S5 Table. Individual infants' Z-scores of deoxy-Hb for cartoon character face faces in each of channels (Experiment 2).**
(TIF)

## Acknowledgments

We thank Jaile Yang, Yuki Tsuji, Yusuke Nakashima, Megumi Kobayashi, Shuma Tsurumi for their assistance with data collection. Special thanks to the infants and their parents for their kindness and cooperation.

## Author Contributions

**Conceptualization:** Nanako Yamanaka, So Kanazawa, Masami K. Yamaguchi.

**Data curation:** Nanako Yamanaka.

**Formal analysis:** Nanako Yamanaka.

**Funding acquisition:** Masami K. Yamaguchi.

**Investigation:** Nanako Yamanaka.

**Methodology:** Nanako Yamanaka, So Kanazawa, Masami K. Yamaguchi.

**Project administration:** Masami K. Yamaguchi.

**Supervision:** So Kanazawa, Masami K. Yamaguchi.

**Visualization:** Nanako Yamanaka.

**Writing – original draft:** Nanako Yamanaka.

**Writing – review & editing:** So Kanazawa, Masami K. Yamaguchi.

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
