## [Decision Letter · Decision Letter 0]

14 Oct 2021

PONE-D-21-19037Infants’ brain activity to cartoon face using functional near-infrared spectroscopy‎PLOS ONE

Dear Dr. Yamanaka,

Thank you for submitting your manuscript to PLOS ONE. After careful consideration, we feel that it has merit but does not fully meet PLOS ONE’s publication criteria as it currently stands. Therefore, we invite you to submit a revised version of the manuscript that addresses the points raised during the review process.

Two experts in the field have carefully reviewed the manuscript entitled, "Infants’ brain activity to cartoon face using functional near-infrared spectroscopy‎". Their comments are appended below.

Although both of them acknowledged the findings obtained by this study is interesting,the samaple size is too small to draw the clear-cut result. Furthermos they pointed out the Discussion section should be strengthed.

I am looking forward to receiving the adequate replies to each critiques and the necessary revision.

We look forward to receiving your revised manuscript.

Kind regards,

Manabu Sakakibara, Ph.D.

Academic Editor

PLOS ONE

Journal Requirements:

This research was financially supported by “Construction of the Face-Body Studies in Transcultural Conditions” (17H06343). 

This research was financially supported by “Construction of the Face-Body Studies in Transcultural Conditions” (17H06343).

5. We note that Figure 1 in your submission contain copyrighted images. All PLOS content is published under the Creative Commons Attribution License (CC BY 4.0), which means that the manuscript, images, and Supporting Information files will be freely available online, and any third party is permitted to access, download, copy, distribute, and use these materials in any way, even commercially, with proper attribution. For more information, see our copyright guidelines: http://journals.plos.org/plosone/s/licenses-and-copyright.

6. We noticed you have some minor occurrence of overlapping text with the following previous publication(s), which needs to be addressed:

- https://www.sciencedirect.com/science/article/pii/S1053811918321098?via%3Dihub

-https://www.sciencedirect.com/science/article/abs/pii/S0022096511001858?via%3Dihub

 The text that needs to be addressed involves the Result, Data Analysis, and Introduction sections.

In your revision ensure you cite all your sources (including your own works), and quote or rephrase any duplicated text outside the methods section. Further consideration is dependent on these concerns being addressed.

Reviewers' comments:

Reviewer's Responses to Questions

**Comments to the Author**

1. Is the manuscript technically sound, and do the data support the conclusions?

Reviewer #1: Partly

Reviewer #2: Yes

2. Has the statistical analysis been performed appropriately and rigorously? 

Reviewer #1: No

Reviewer #2: Yes

3. Have the authors made all data underlying the findings in their manuscript fully available?

Reviewer #1: Yes

Reviewer #2: Yes

4. Is the manuscript presented in an intelligible fashion and written in standard English?

Reviewer #1: Yes

Reviewer #2: No

5. Review Comments to the Author

Reviewer #1: Comments:

A novel study presented in this manuscript however, many details are unclear, and the manuscript needs improvement. The results presented are interesting however, the sample size is very small.

Data analysis, Line 273:

It’s unclear what the authors mean by “raw oxy-Hb signal”. Where the changes in concentration calculated from the attenuation change data? If so, how was this done? Were in-house scripts used? What was the procedure? The use of just a band-pass filter seems highly minimal – did the authors not use any type of motion correction for the data as infant data can have many motion artifacts?

Could the authors also report the deoxy-Hb signal results? It is important to look at both the changes in oxy-Hb and deoxy-Hb.

The authors need to improve the data analysis section in Experiment 2 to reflect the points above and the deoxy-Hb results need to be added in. As it stands, the data analysis section is confusing and it’s unclear what exactly is being shown. It’s also unclear why the authors chose to present “z-score” of the “raw signal” rather than changes in concentration.

Result, Line 315:

I believe that the statistical testing could be performed in a more accurate way. For example, rather than testing against 0, the mean z-score should be tested against the mean z-score in the baseline period. This would give a more accurate statistical comparison. Could the authors please repeat the statistical analysis?

Discussion

The work presented in this paper seems novel in that this specific effect has not been studied in infants before this study. However, besides restating the results, the authors don’t offer any discussion or explanation of their results. For example, Channel 23 was the only channel that showed significant activation – do the authors have a hypothesis about this? Why is this channel specifically important for these stimuli? Most of the discussion section reads as a literature review with lots of studies cited. The authors need to work on the discussion and weave in the studies to support their hypotheses and results as opposed to how it is currently written.

Reviewer #2: The authors investigated two face-stimuli-related experiments for 5- and 6-month-old infants. The behavioral one showed a preference for cartoon female faces in the upright, while another NIRS one showed that the concentration of oxy-Hb increased in the case of cartoon female faces. These results are, in general, interesting for readers. Some comments and suggestions should be considered in a modified version of the manuscript.

1. For any subject, the authors should obtain both behavioral measure data and NIRS imaging data simultaneously. Then, they should detect if there is correlative between behavioral and NIRS measurement.

2. The sample size is relatively small and there is a significant gender difference of sample. The authors should increase more samples in their study if possible.

3. The explanation of their results should be strengthened. More importantly, the mechanism underlying the current study should be discussed more deeply. In addition, it is expected to know if the current results can also be extended to infants with high risk of developmental disorder.

6. PLOS authors have the option to publish the peer review history of their article (what does this mean?). If published, this will include your full peer review and any attached files.

Reviewer #1: No

Reviewer #2: No

---

## [Author Response · Author response to Decision Letter 0]

26 Nov 2021

Reviewer #1: We have incorporated all of your comments into my revised manuscript. Thank you for your help.

Reviewer #2: We have incorporated all of your suggestions into my revision. Thank you for your comments.

---

## [Decision Letter · Decision Letter 1]

3 Jan 2022

Infants’ brain activity to cartoon face using functional near-infrared spectroscopy‎

PONE-D-21-19037R1

Dear Dr. Yamanaka,

The original reviewer #1 and this Academic Editor carefully reviewed the revision.  The external reviewer is satisfied with the revised manuscript, and I judged the revision is scientifically acceptable for publication in PLOS ONE.

We’re pleased to inform you that your manuscript has been judged scientifically suitable for publication and will be formally accepted for publication once it meets all outstanding technical requirements.

Kind regards,

Manabu Sakakibara, Ph.D.

Academic Editor

PLOS ONE

Additional Editor Comments (optional):

Reviewers' comments:

Reviewer's Responses to Questions

**Comments to the Author**

1. If the authors have adequately addressed your comments raised in a previous round of review and you feel that this manuscript is now acceptable for publication, you may indicate that here to bypass the “Comments to the Author” section, enter your conflict of interest statement in the “Confidential to Editor” section, and submit your "Accept" recommendation.

Reviewer #1: All comments have been addressed

2. Is the manuscript technically sound, and do the data support the conclusions?

Reviewer #1: Yes

3. Has the statistical analysis been performed appropriately and rigorously? 

Reviewer #1: Yes

4. Have the authors made all data underlying the findings in their manuscript fully available?

Reviewer #1: Yes

5. Is the manuscript presented in an intelligible fashion and written in standard English?

Reviewer #1: Yes

6. Review Comments to the Author

Reviewer #1: The authors have addressed all the points raised during the first review. I believe the manuscript is in a form to be accepted for publication.

7. PLOS authors have the option to publish the peer review history of their article (what does this mean?). If published, this will include your full peer review and any attached files.

Reviewer #1: No

---

## [Editor Report · Acceptance letter]

25 Jan 2022

PONE-D-21-19037R1 

Infants’ brain activity to cartoon face using functional near-infrared spectroscopy 

Dear Dr. Yamanaka:

I'm pleased to inform you that your manuscript has been deemed suitable for publication in PLOS ONE. Congratulations! Your manuscript is now with our production department. 

Kind regards, 

on behalf of

Dr. Manabu Sakakibara 

Academic Editor

PLOS ONE